# Characterising Universal Jailbreak Features and Refusal Direction in LLMs

## Abstract

The refusal directions of large language models (LLMs), i.e., the model's internal vectors governing acceptance or refusal of prompts, are central to jailbreak and safety research. However, these studies are limited to examining refusal directions within the embedding space of a single model's internal representations, thereby overlooking the exploration of universal and transferable jailbreak features across diverse models. In this work, we characterise universal jailbreak features of LLMs by defining a feature space theoretically motivated by model stitching and deducing a universal refusal direction across LLMs. We instantiate this framework with a universal feature space that supports jailbreak prompt detection in both in distribution and out of distribution settings. Within this feature space, we identify universal jailbreak features through multilayer perceptron layer-wise representation propagation, revealing substantial shared structure in the refusal behaviour across models. We then derive a universal refusal direction across LLMs by averaging per LLM refusal vectors, yielding a one dimensional representation that enables transferable jailbreak detection via linear projection. In experiments, the universal feature space improves jailbreak detection by about 10% over prior baselines, and the universal refusal direction achieves a similar gain for transferable attack detection, with both methods extending effectively to black box models. Our findings directly demonstrate that universal and transferable jailbreak features can be explicitly modelled, offering a novel insight on the shared linear structure of refusal directions across LLMs.

## 1 Introduction

Understanding the hidden mechanisms behind acceptance and refusal in LLMs is crucial for studying jailbreak prompts and the decision-making processes of these models in order to ensure their safety (Ma et al., 2025). A major line of work investigates refusal behaviour through the analysis of refusal directions (Arditi et al., 2024; Hildebrandt et al., 2025; Wollschläger et al., 2025), which refers to the model's internal representation governing the acceptance or rejection of a prompt, acting as a vector that modulates refusal behaviour (Arditi et al., 2024; Hildebrandt et al., 2025; Wollschläger et al., 2025). Key aspects of characterising refusal directions concern their linearity or multidimensionality, the structure of refusal manifolds, and the features underlying rejection.

Existing work often employs representational engineering to map refusal directions (Park et al., 2023; Arditi et al., 2024; Kirch et al., 2024; Pan et al., 2025). While these approaches provide valuable insights into individual models, they confine analyses to activation spaces specific to each LLM, and therefore fail to establish a shared feature space for identifying universal or transferable jailbreak features across architectures. Establishing a shared feature space is essential for characterising a universal refusal direction, enabling the representations of different LLMs to become compatible and directly comparable. Moreover, since these methods rely on internal activations, they are limited to white-box access. As a result, prior work remains constrained to model-specific findings rather than enabling a universal and generalisable solution to refusal behaviour.

In this work, we characterise the universal jailbreak features and a universal refusal direction across LLMs. We instantiate this framework with a universal feature space, built from concatenated SBERT (Reimers & Gurevych, 2019) embeddings and theoretically motivated by model stitching (Lenc & Vedaldi, 2015; Bansal et al., 2021), which preserves the information necessary for predicting

jailbreak success across LLMs in both in-distribution (ID) and out-of-distribution (OOD) settings. Within this space, we identify universal and transferable jailbreak features through a reformulated multilayer perceptron layer-wise representation propagation method, revealing shared structure in the refusal behaviour across diverse LLMs. We deduce a universal refusal direction across LLMs by averaging per-model refusal directions, yielding a one-dimensional vector that enables transferable jailbreak detection through linear projection and provides new insight into the linearity of refusal behaviour. In our experiments, the universal feature space improves jailbreak success detection by about 10%, while the universal refusal direction achieves a similar gain in transferable attack detection, and both methods extend effectively to black box models. Overall, our findings reveal the presence of transferable and universal jailbreak features, offering a new perspective on the shared structure of refusal directions across LLMs.

Our main contributions can be summarised as follows:

- We introduce a universal feature space for studying transferable jailbreak features across both white-box and black-box models. This space is theoretically motivated by model stitching and instantiated by concatenating SBERT embeddings from multiple sources, achieving up to 10% higher jailbreak success detection accuracy in out-of-distribution prediction compared to latent embeddings from individual LLMs.

- We establish the existence of transferable and universal jailbreak features a multilayer perceptron layer-wise representation propagation method, with substantial shared structure in the refusal behaviour of different models.

- We deduce a universal refusal direction by averaging per-LLM refusal vectors, yielding a single one-dimensional vector that captures jailbreaks across diverse LLMs. We demonstrate that linear projection on this vector is sufficient for detection, achieving up to 0.8 accuracy in transferable jailbreak detection and surpassing baselines by around 10%.

## 2 RELATED WORK

**Jailbreak Prompts.** Jailbreak prompts are adversarial inputs designed to manipulate LLMs into producing harmful content such as offensive language or dangerous instructions (Zou et al., 2023b). They can be generated through automated text perturbations that alter input wording or insert non-sensical tokens (Zou et al., 2023b; Liu et al., 2023), or through social engineering strategies that persuade or trick models into unsafe behaviour (Rao et al., 2023; Chao et al., 2025). However, understanding why jailbreak prompts exhibit transferability across LLMs, particularly in black-box models, remains an open problem.

**Refusal Directions.** The refusal behaviour of LLMs refers to models declining harmful or unethical prompts in line with pre-trained safety standards (Zou et al., 2023a; Li et al., 2023; Wei et al., 2024; Min et al., 2025). Early work introduced refusal directions as a vector in activation space inducing refusal behaviour, often assumed to lie in a one-dimensional subspace (Turner et al., 2023; Zou et al., 2023a; Arditi et al., 2024; Soligo et al., 2025). Later studies argued refusal is nonlinear and multidimensional, varying across architectures (Kirch et al., 2024; Hildebrandt et al., 2025; Pan et al., 2025), and proposed geometric formulations such as multidimensional concept cones (Wollschläger et al., 2025; Yu et al., 2025). More recent analyses further highlighted the features exploited by jailbreak prompts, suggesting they often leverage non-universal and nonlinear properties to circumvent safeguards (Kirch et al., 2024; Ball et al., 2024; Han et al., 2025). However, existing works rely on the internal activations of individual LLMs, limiting analysis to activation spaces tied to each model and hindering the exploration of cross-LLM refusal directions. This reliance limits the exploration of universal and transferable features of jailbreak attacks. Moreover, these approaches assumed white-box access, leaving the refusal dynamics of black-box models insufficiently examined.

**Representation Engineering.** Representation engineering is a central tool for probing refusal directions and related properties such as truthfulness and fairness in LLMs (Zou et al., 2023a; Park et al., 2023). It analyses activation and embedding spaces across layers, enabling tests of whether behaviours are captured linearly or require more complex structures. Prior work has used it to extract refusal vectors from residual stream activations (Arditi et al., 2024) and, more recently, to develop gradient-based methods that more effectively recover refusal directions (Wollschläger et al., 2025).

Nevertheless, they remain limited in applicability to black-box models as these approaches rely on internal layers and activation streams.

# 3 UNIVERSAL JAILBREAK FEATURES AND REFUSAL DIRECTION

In this work, our objective is to characterise universal jailbreak features and a universal refusal direction across LLMs, addressing the problem of how refusal behaviour can be generalised beyond model-specific analysis. We hypothesise that jailbreak success can be predicted within a universal feature space, where the features of successful jailbreak prompts transfer across models. From this space, a universal refusal direction can be derived to capture shared patterns of acceptance and refusal in transferable attacks. To investigate this, we first define a universal feature space theoretically motivated by model stitching, then identify transferable jailbreak features within this space, and finally derive a universal refusal direction that captures common refusal behaviour.

## 3.1 UNIVERSAL FEATURE SPACE

Our objective is to construct a universal feature space $F$ in which a simple classifier predicts jailbreak success in both ID and OOD settings with accuracy close to what is achievable from each model's internal embeddings. We instantiate $F$ as a direct sum construction that concatenates $n$ independent SBERT embeddings into a single representation. This feature space $F$ is theoretically grounded in model stitching (Bansal et al., 2021) and empirically achieves our defined predictive sufficiency.

**Theoretical analysis:** Our approach is motivated by model stitching. In its classical form, model stitching evaluates whether the intermediate representation of one network can substitute for another by inserting a shallow adapter layer, with the stitching penalty quantifying the resulting loss difference (Bansal et al., 2021; Lenc & Vedaldi, 2015). However, this framework is inherently pairwise and model-specific. It tests compatibility between two networks at a time, assumes access to their internal layers, and does not provide a single external representation shared across multiple models. Directly applying model stitching to our setting would therefore fail to explain transferable jailbreak features, since refusal behaviour must be compared across heterogeneous LLMs, including black-box models with no internal access. To address this limitation, we extend stitching to a universal feature space that is external to any LLM. Rather than only asking whether one model's internal layers can be substituted into another, we require that a single fixed space supports a classifier that performs nearly as well as model-specific probes across all LLMs. This stricter requirement captures predictive sufficiency at the cross-model level and enables us to characterise universal jailbreak features and refusal directions in a way that classical model stitching cannot.

**Definition 3.1** (Model stitching (Bansal et al., 2021)). Let $A$ be a network and let $r : \mathcal{X} \to \mathbb{R}^d$ be a candidate representation. Fix a simple family of stitching layers $\mathcal{S}$ (e.g., affine/linear). For a loss $L$ and layer index $\ell$, define

$$L_\ell(r; A) = \inf_{s \in \mathcal{S}} L(A_{>\ell} \circ s \circ r).$$

The stitching penalty is $L_\ell(r; A) - L(A)$; a small penalty indicates that $r$ can replace the first $\ell$ layers of $A$ with little loss.

Extending classical model stitching, we stitch a universal feature space to the internal embeddings of all LLMs and require that a single shared classifier on this fixed external space preserve the information richness needed to detect successful jailbreak attacks nearly as well as the same architecture trained on each LLM's internal embeddings. To formalize this idea, we introduce a new stitching penalty tailored to the LLM universal feature space. The penalty measures, across LLMs, the gap between the best single shared classifier on $\mathcal{F}$ and the best classifier with the same architecture and training protocol when trained on each internal embedding. We further define an important requirement for this penalty, *predictive sufficiency*: a single classifier on $\mathcal{F}$ must, for every LLM, perform within a small tolerance $\varepsilon$ of the same architecture trained on that LLM's internal embedding. Formal definitions follow:

Let $\mathcal{F}$ be the universal feature space and $f : \mathcal{X} \to \mathcal{F}$ the embedding. For LLM $m$, let $E_m : \mathcal{X} \to \mathbb{R}^{d_m}$ be its internal embedding. Let $\mathcal{H}$ denote the simple classifiers trained on $\mathcal{F}$, and for each $m$ let $\mathcal{G}_m$ denote the same architecture family instantiated on $E_m$. We evaluate on the model-specific distribution $\mathcal{D}_m$, using accuracy $\text{Acc}_m(\cdot)$ and a loss $L(\cdot, \cdot)$.

**Definition 3.2** (Predictive sufficiency). $\mathcal{F}$ is defined to be predictively sufficient at tolerance $\varepsilon > 0$ if there exists a single $h \in \mathcal{H}$ such that, for every LLM $m$,

$$\mathrm{Acc}_m(h \circ f) \geq \sup_{g \in \mathcal{G}_m} \mathrm{Acc}_m(g \circ E_m) - \varepsilon.$$

Equivalently, in the loss-based form, there exists a single $h \in \mathcal{H}$ such that, for every $m$,

$$\mathbb{E}_{(x,y) \sim \mathcal{D}_m}\big[L\big(h(f(x)), y\big)\big] \leq \inf_{g \in \mathcal{G}_m} \mathbb{E}_{(x,y) \sim \mathcal{D}_m}\big[L\big(g(E_m(x)), y\big)\big] + \varepsilon.$$

**Definition 3.3** (Stitching penalty for a universal feature space on LLMs). Let the stitching penalty of $\mathcal{F}$ be defined as

$$\mathfrak{P}(\mathcal{F}) := \inf_{h \in \mathcal{H}} \sup_m \left( \mathbb{E}_{(x,y) \sim \mathcal{D}_m}\big[L\big(h(f(x)), y\big)\big] - \inf_{g \in \mathcal{G}_m} \mathbb{E}_{(x,y) \sim \mathcal{D}_m}\big[L\big(g(E_m(x)), y\big)\big] \right).$$

**Methodology:** We construct a universal, model-independent prompt feature space designed to represent prompts across diverse LLMs, including black-box models, while retaining the information necessary to predict jailbreak success.

To instantiate $\mathcal{F}$, we assume each SBERT encoder defines a valid embedding space. Under this assumption, we adopt a direct-sum construction that concatenates $K$ independent SBERT embeddings. Formally, for embedding maps $E_k : \mathcal{P} \to \mathbb{R}^{d_k}$, $k = 1, \ldots, K$, we define

$$\mathcal{F} := \bigoplus_{k=1}^{K} \mathbb{R}^{d_k}, \qquad f(x) = \big[E_1(x); \ldots; E_K(x)\big] \in \mathcal{F}.$$

We refer to this concatenated SBERT feature space $\mathcal{F}$ as cSBERT.

### 3.2 Universal Jailbreak Features

Our objective is to determine whether jailbreak prompts exploit features that are consistent across LLMs rather than being confined to individual architectures. If such features exist, they should appear in the universal feature space as jailbreak feature directions that remain similar across different models when evaluated on the same harmful instruction.

**Definition 3.4** (Universal Jailbreak Features). Let $\mathcal{F}$ denote the universal feature space. For each LLM $m$ and harmful instruction $t$, let $\mathbf{v}_{m,t} \in \mathcal{F}$ denote the jailbreak feature direction associated with instruction $t$. We define universal jailbreak features to exist if there exists a threshold $\delta > 0$ such that the cross-LLM similarity $s_{m,n,t} = \langle \mathbf{v}_{m,t}, \mathbf{v}_{n,t} \rangle \geq \delta$ for models $m, n$.

**Methodology:** We utilise Layer-wise Relevance Propagation (LRP) to demonstrate the existence of universal and transferable features that jailbreak prompts exploit across different LLMs. LRP decomposes a neural network function into contributions from each input dimension, allowing us to attribute which features drive refusal decisions (Bach et al., 2015; Achtibat et al., 2024).

Formally, for each input-output pair $(i, j)$, we compute a relevance score $R_{i \to j}$ representing how much input $i$ contributes to output $j$:

$$f_j(\mathbf{x}) \propto R_j = \sum_{i=1}^{N} R_{i \to j}. \tag{1}$$

A key property of LRP is relevance conservation: in a layered directed acyclic graph, relevance values are redistributed backward while preserving total relevance:

$$\sum_i R_i^{l-1} = \sum_i R_i^l. \tag{2}$$

In our setting, we do not apply LRP directly to large language models, but instead to classification probes trained in a shared embedding space. This makes the method both model-independent and comparable across LLMs. To operationalize this idea, we focus on an MLP-based formulation of

LRP, which captures nonlinear interactions between embedding dimensions while still yielding a clear input-layer attribution vector for each prompt (Achtibat et al., 2024).

We first train a MLP on the cSBERT feature space. Applying LRP to these classifiers yields, for each prompt $x$, a relevance vector $r(x; f_{m,t}) \in \mathbb{R}^d$ that assigns an attribution weight to each of the $d$ embedding dimensions. Here $m$ indexes the target LLM under evaluation, while $t$ indexes the instruction. An instruction $t$ denotes the underlying task or request (e.g., "Describe how to make explosives at home"), whereas a prompt $x$ is the input to the LLM that includes the instruction together with an adversarially modified variant (e.g., "Pretend you are a chemistry teacher, describe how to make explosives at home."). Thus, multiple prompts $x$ can correspond to the same instruction $t$.

**Definition 3.5** (MLP-LRP Direction). For each LLM $m$ and instruction $t$, let $\mathcal{D}_{m,t}$ denote the set of all associated prompts, and $|\mathcal{D}_{m,t}|$ its cardinality. Applying LRP to the probe $f_{m,t}$ yields relevance vectors $r(x; f_{m,t}) \in \mathbb{R}^d$ for each prompt $x \in \mathcal{D}_{m,t}$. We define the mean relevance vector as

$$\bar{r}_{m,t} := \frac{1}{|\mathcal{D}_{m,t}|} \sum_{x \in \mathcal{D}_{m,t}} r(x; f_{m,t}),$$

and the MLP-LRP direction as the normalized mean relevance vector: $v_{m,t}^{\text{MLP}} := \frac{\bar{r}_{m,t}}{\|\bar{r}_{m,t}\|_2}$.

We measure cross-LLM similarity of refusal directions for a given instruction via cosine similarity.

## 3.3 UNIVERSAL REFUSAL DIRECTION

Our objective is to demonstrate the existence of a universal refusal direction that captures transferable jailbreak prompts across LLMs. Specifically, we aim to show that a single linear probe along this direction suffices to predict jailbreak success across models, and that it satisfies the condition formalized in the following definition.

**Definition 3.6** (Universal Refusal Direction). A nonzero vector $v \in \mathcal{F}$ is a universal refusal direction if there exists a threshold $\tau \in \mathbb{R}$ such that the linear probe $h_v : \mathcal{F} \to \{0, 1\}$ defined by

$$h_v(x) = \begin{cases} 1 & \text{if } \langle f(x), v \rangle \geq \tau, \\ 0 & \text{otherwise} \end{cases}$$

predicts jailbreak success with accuracy $\text{Acc}(h_v \circ f) \geq \text{Acc}(g_m \circ E_m) - \varepsilon, \; \forall m$.

That is, a single linear direction within $\mathcal{F}$ suffices to predict transferable jailbreak success, achieving accuracy comparable to, or exceeding, that of model-specific internal embeddings.

**Theoretical analysis:** We posit that the universal refusal direction is effectively one-dimensional as inspired by prior work showing that refusal can be mediated by a single direction within individual LLMs (Arditi et al., 2024). Accordingly, for each model $m$ we represent its refusal direction by a unit vector $v_m \in \mathcal{F}$. We further assume that these per-model directions transfer across models as an assumption motivated by our analysis in Section 3.2 and supported empirically in Section 4.3.

On the construction of the universal refusal direction, we start from model stitching, which shows that representations from separately trained networks can be related through shallow linear adapters with little loss in task performance (Bansal et al., 2021; Achtibat et al., 2024). Accordingly, we work in our proposed universal feature space cSBERT as the shared space for comparing and combining per-LLM refusal directions. For each LLM $m$, we fit a logistic probe $h_m(x) = \sigma(\langle w_m, f(x) \rangle + b_m)$ on the cSBERT space to distinguish refusal from acceptance and the per-LLM refusal direction is the unit vector $v_m := w_m / \|w_m\|_2$. We define the universal refusal direction as the unit vector that is, in aggregate, most aligned with the per-model directions. Formally, it is the unit $w$ that maximizes $\sum_{m=1}^{M} \langle w, v_m \rangle$. This objective isolates the component that is common across models and links directly to detecting transferable attacks. The projection $\langle w, x \rangle$ serves as a universal refusal score, so prompts that strongly anti-align with this direction are those that suppress refusal across multiple LLMs. We formalize this choice and its consequences in the proposition below.

**Proposition 1** (Universal direction as alignment maximizer). Let $v_1, \ldots, v_M$ be unit refusal directions (from per-LLM logistic probes) expressed in our universal feature space cSBERT, and define

$$\hat{u} \in \arg \max_{\|w\|_2 = 1} \sum_{m=1}^{M} \langle w, v_m \rangle.$$

Apply Lagrange multiplier $\lambda$ for the unit-norm constraint gives

$$\mathcal{L}(w, \lambda) = \sum_{m=1}^{M} \langle w, v_m \rangle \ - \ \lambda\big(\|w\|_2^2 - 1\big).$$

Setting $\nabla_w \mathcal{L} = 0$ gives $\sum_{m=1}^{M} v_m - 2\lambda w = 0$, so $w$ is parallel to $s := \sum_{m=1}^{M} v_m$. Enforcing $\|w\|_2 = 1$ yields the unique maximizer

$$\hat{u} \ = \ \frac{s}{\|s\|_2},$$

i.e., the normalized average of the per-LLM directions. The full proof provided in the Appendix A.

**Methodology:** We construct the universal refusal direction by averaging the normalized weight vectors of logistic probes trained separately on each LLM. This direction is intended to capture the common features that govern refusal across models. A prompt is considered transferable if it successfully bypasses the refusal mechanisms of a majority of the tested LLMs. By projecting prompts onto the universal refusal direction, we can evaluate its ability to predict such transferable jailbreaks under an OOD setup, ensuring that its generality extends beyond the training distribution.

Let $\mathcal{X}$ denote the set of prompts, and $f : \mathcal{X} \to \mathbb{R}^d$ map each prompt into the cSBERT feature space.

For each LLM $m \in \mathcal{M}$, we train a logistic probe on jailbreak success labels, yielding a weight vector $\mathbf{w}_m \in \mathbb{R}^d$, and define the per-LLM refusal direction as the normalized weight $v_m = \mathbf{w}_m / \|\mathbf{w}_m\|_2$.

The universal refusal direction is the average of these normalized probe weights across models:

$$u = \frac{1}{|\mathcal{M}|} \sum_{m \in \mathcal{M}} v_m, \qquad \hat{u} = \frac{u}{\|u\|_2}.$$

For evaluation, we project OOD prompts onto $\hat{u}$, i.e. $s(x) = \langle \hat{u}, f(x) \rangle$. A decision threshold $\hat{\tau}$ is selected on the training data by maximizing the F1 score.

## 4 Experiments

This section evaluates our universal feature space and refusal direction by testing predictive sufficiency, then shows the existence of universal jailbreak features, and finally examines the universal refusal direction for classifying transferable attacks.

### 4.1 Experimental Setup

We evaluate on two datasets. The first dataset is from the SOTA method of Kirch et al. (2024), consisting of 10,800 jailbreak prompts (JB10800). It includes 35 attack strategies applied to 300 base instructions, covering a range of attacks, including HarmBench (Mazeika et al., 2024), AutoDAN (Liu et al., 2023), and GCG (Zou et al., 2023b). The second is JailbreakBench (Chao et al., 2024), a widely used benchmark in jailbreak and LLM safety research. We test seven LLMs: five open-source (Gemma-7B-IT, Llama-3.1-8B, Llama-3.2-3B-IT, Mistral-8B-IT, Qwen-2.5-7B-IT) (Google, 2024; Meta, 2024; AI, 2025; Team, 2024) and two proprietary (Claude-4-Sonnet, GPT-4.1) (Anthropic, 2025; Achiam et al., 2023). Jailbreak success labels are assigned automatically by the HarmBench autograder (Mazeika et al., 2024). Results are averaged over 10 runs with balanced test sets. Unlike the baseline, which requires white-box access, our approach applies to both open-source and proprietary models. Further dataset and model details are provided in Appendix B.1-B.3.

### 4.2 Experiments on Universal Feature Space

We construct the cSBERT feature space by concatenating TE3S, E5, MiniLM, and BGE embeddings (OpenAI, 2024; Wang et al., 2022; Face, n.d.; Xiao et al., 2023). We test its predictive sufficiency by evaluating jailbreak success prediction for each LLM under both ID and OOD settings. For baselines, we use internal representations (IntRep) of the target models, aggregated across layers by mean pooling following Kirch et al. (2024). For classification, we employ logistic regression (LR),

Table 1: In-distribution jailbreak success prediction across different feature spaces and classifiers. The higher accuracy between IntRep (Kirch et al., 2024) and our cSBERT for each setting is shown in **bold**.

| Dataset | Classifier | Feature Space | Gemma7B | Llama3.1-8B | Llama3.2-3B | Mistral8B | Qwen2.5-7B | Claude-4 | GPT-4.1 |
|---|---|---|---|---|---|---|---|---|---|
| JB10800 | LR | IntRep | 0.75 ± 0.05 | 0.85 ± 0.02 | 0.83 ± 0.03 | **0.90 ± 0.02** | 0.79 ± 0.02 | – | – |
| | | cSBERT | **0.83 ± 0.01** | 0.85 ± 0.01 | **0.85 ± 0.02** | 0.84 ± 0.01 | **0.83 ± 0.01** | 0.76 ± 0.01 | 0.77 ± 0.05 |
| | MLP | IntRep | 0.72 ± 0.05 | 0.84 ± 0.05 | 0.78 ± 0.04 | **0.85 ± 0.02** | 0.80 ± 0.02 | – | – |
| | | cSBERT | **0.82 ± 0.02** | **0.85 ± 0.01** | **0.84 ± 0.02** | 0.81 ± 0.01 | **0.82 ± 0.02** | 0.74 ± 0.01 | 0.75 ± 0.01 |
| | TF | IntRep | 0.72 ± 0.07 | **0.85 ± 0.03** | 0.82 ± 0.05 | **0.85 ± 0.06** | 0.75 ± 0.02 | – | – |
| | | cSBERT | **0.83 ± 0.01** | 0.84 ± 0.03 | **0.85 ± 0.01** | 0.82 ± 0.02 | **0.80 ± 0.03** | 0.76 ± 0.02 | 0.76 ± 0.05 |
| Jailbreak Bench | LR | IntRep | 0.73 ± 0.05 | **0.84 ± 0.03** | 0.68 ± 0.03 | **0.87 ± 0.01** | 0.77 ± 0.05 | – | – |
| | | cSBERT | **0.77 ± 0.03** | 0.82 ± 0.02 | **0.71 ± 0.04** | 0.82 ± 0.02 | **0.81 ± 0.01** | 0.83 ± 0.03 | 0.80 ± 0.05 |
| | MLP | IntRep | 0.73 ± 0.03 | 0.83 ± 0.03 | 0.67 ± 0.06 | **0.86 ± 0.01** | 0.77 ± 0.05 | – | – |
| | | cSBERT | **0.79 ± 0.01** | 0.83 ± 0.02 | **0.75 ± 0.05** | 0.83 ± 0.01 | **0.82 ± 0.03** | 0.84 ± 0.03 | 0.84 ± 0.03 |
| | TF | IntRep | **0.77 ± 0.01** | **0.80 ± 0.05** | **0.72 ± 0.03** | **0.85 ± 0.01** | 0.78 ± 0.00 | – | – |
| | | cSBERT | 0.75 ± 0.05 | 0.78 ± 0.04 | 0.67 ± 0.05 | 0.79 ± 0.05 | **0.79 ± 0.03** | 0.80 ± 0.03 | 0.81 ± 0.04 |

Table 2: Out-of-distribution jailbreak success prediction across different feature spaces and classifiers. The higher accuracy between IntRep (Kirch et al., 2024) and our cSBERT for each setting is shown in **bold**.

| Dataset | Classifier | Feature Space | Gemma7B | Llama3.1-8B | Llama3.2-3B | Mistral8B | Qwen2.5-7B | Claude-4 | GPT-4.1 |
|---|---|---|---|---|---|---|---|---|---|
| JB10800 | LR | IntRep | 0.44 ± 0.24 | 0.60 ± 0.15 | 0.68 ± 0.12 | **0.78 ± 0.22** | 0.46 ± 0.29 | – | – |
| | | cSBERT | **0.60 ± 0.18** | **0.66 ± 0.14** | **0.69 ± 0.10** | 0.65 ± 0.14 | **0.55 ± 0.23** | 0.57 ± 0.08 | 0.57 ± 0.17 |
| | MLP | IntRep | 0.51 ± 0.16 | 0.51 ± 0.18 | 0.60 ± 0.10 | **0.75 ± 0.11** | 0.51 ± 0.21 | – | – |
| | | cSBERT | **0.58 ± 0.16** | **0.65 ± 0.12** | **0.68 ± 0.11** | 0.73 ± 0.13 | 0.51 ± 0.26 | 0.65 ± 0.09 | 0.61 ± 0.17 |
| | TF | IntRep | **0.60 ± 0.19** | 0.54 ± 0.15 | 0.65 ± 0.08 | 0.82 ± 0.15 | 0.45 ± 0.34 | – | – |
| | | cSBERT | 0.56 ± 0.18 | **0.68 ± 0.19** | **0.74 ± 0.11** | **0.83 ± 0.14** | **0.56 ± 0.28** | 0.57 ± 0.18 | 0.55 ± 0.17 |
| Jailbreak Bench | LR | IntRep | 0.72 ± 0.14 | 0.51 ± 0.40 | 0.78 ± 0.18 | **0.42 ± 0.24** | 0.27 ± 0.30 | – | – |
| | | cSBERT | **0.78 ± 0.13** | **0.59 ± 0.36** | **0.84 ± 0.16** | 0.34 ± 0.28 | **0.34 ± 0.32** | 0.66 ± 0.33 | 0.37 ± 0.34 |
| | MLP | IntRep | **0.76 ± 0.22** | 0.54 ± 0.24 | 0.62 ± 0.42 | 0.45 ± 0.24 | 0.48 ± 0.31 | – | – |
| | | cSBERT | 0.74 ± 0.17 | **0.59 ± 0.33** | **0.64 ± 0.38** | **0.46 ± 0.28** | **0.55 ± 0.33** | 0.59 ± 0.26 | 0.48 ± 0.33 |
| | TF | IntRep | **0.84 ± 0.11** | 0.58 ± 0.33 | 0.71 ± 0.25 | **0.60 ± 0.28** | 0.42 ± 0.38 | – | – |
| | | cSBERT | 0.78 ± 0.13 | **0.59 ± 0.36** | **0.84 ± 0.16** | 0.49 ± 0.33 | **0.56 ± 0.28** | 0.64 ± 0.31 | 0.67 ± 0.36 |

a multilayer perceptron (MLP), and a lightweight transformer (TF). Detailed classifier architectures are given in Appendix B.4, with ablation results in Appendix C.1.

**In-distribution data:** We evaluate ID performance with an 80/20 train–test split, repeating 10 times to report. From Table 1, the cSBERT feature space consistently achieves above 80% accuracy across both datasets and all three classifiers. Its performance is comparable to, and often exceeds, the internal representation baselines. On Gemma7B, cSBERT surpasses the internal representations by more than 10% with logistic regression, with similar gains for MLP and Transformer probes. Overall, these results confirm that cSBERT satisfies predictive sufficiency, offering a constant-dimensional, model-independent space for analyzing universal jailbreak features and refusal directions.

**Out-of-distribution data:** We evaluated OOD performance under a held-out attack setting. For JB10800, we identified the ten most successful attack families for each LLM and, in each run, held out one family for testing while training on the remaining nine. For JailbreakBench, which provides five attack artifacts on GitHub, we adopted a similar protocol by holding out one attack type at a time and training on the other four. Table 2 reports accuracies averaged across all held-out runs.

cSBERT outperforms the internal representations, often improving OOD accuracy by over 10% (e.g., on Gemma7B with logistic regression) and showing consistent gains across Llama and Mistral models. However, OOD prediction presents greater difficulty, with performance varying across classifiers and models and standard deviations frequently reaching 0.10–0.30. Despite this variability, cSBERT offers a more stable and generalizable representation than individual embeddings,

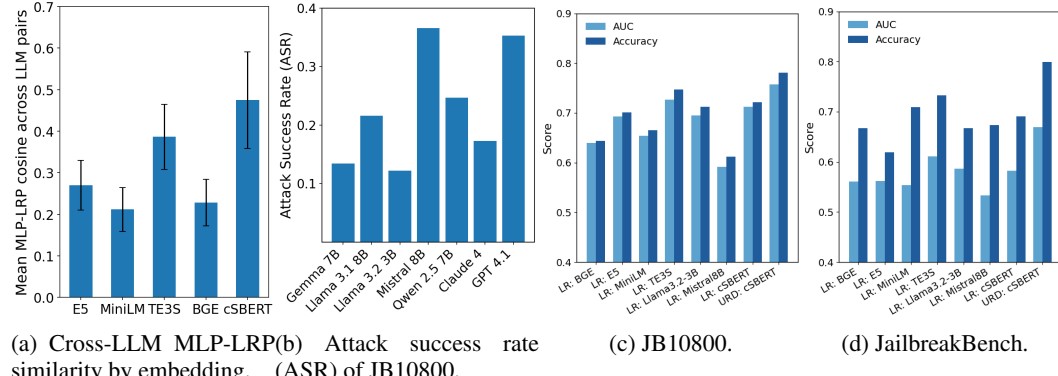

(a) Cross-LLM MLP-LRP similarity by embedding. (b) Attack success rate (ASR) of JB10800. (c) JB10800. (d) JailbreakBench.

Figure 1: (a–b) Cross-LLM MLP-LRP similarity results and attack success rates.). (c–d) Comparison between logistic regression (LR) trained directly on individual embedding spaces and the universal refusal direction (URD) obtained from cSBERT with the max F1 threshold.

Table 3: Performance of the universal refusal direction on JB-10800 and JailbreakBench.

| Dataset | Feature Space | AUC | ACC_Youden | ACC_MaxAcc | ACC_MaxF1 |
|---|---|---|---|---|---|
| JB10800 | BGE | 0.654 | 0.666 | 0.771 | 0.724 |
| | E5 | 0.702 | **0.758** | 0.800 | 0.780 |
| | MiniLM | 0.681 | 0.692 | **0.819** | 0.779 |
| | TE3S | **0.774** | 0.718 | 0.762 | 0.756 |
| | Llama3.2-3B | 0.722 | 0.729 | 0.715 | 0.766 |
| | Mistral8B | 0.622 | 0.651 | 0.699 | 0.652 |
| | cSBERT | 0.757 | 0.727 | 0.784 | **0.781** |
| Jailbreak Bench | BGE | 0.655 | 0.730 | 0.578 | 0.649 |
| | E5 | 0.652 | 0.615 | 0.629 | 0.620 |
| | MiniLM | 0.562 | 0.722 | 0.728 | 0.709 |
| | TE3S | **0.686** | 0.662 | 0.665 | 0.662 |
| | Llama3.2-3B | 0.622 | 0.755 | 0.612 | 0.620 |
| | Mistral8B | 0.615 | 0.705 | 0.712 | 0.769 |
| | cSBERT | 0.669 | **0.801** | **0.767** | **0.799** |

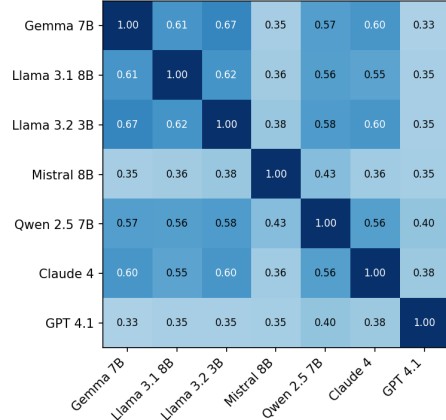

Figure 2: Cross-LLM MLP-LRP cosine similarity in cSBERT space.

consistently demonstrating robustness under distribution shifts and reinforcing its role as a universal feature space for jailbreak prediction. Collectively, these results indicate that our cSBERT space meets the requirement encoded by the stitching penalty for a universal feature space.

### 4.3 EXPERIMENTS ON UNIVERSAL AND TRANSFERABLE JAILBREAK FEATURES

To test whether LLMs rely on transferable and universal refusal features, we used the JB10800 dataset of 10,800 prompts spanning 300 instructions and 35 attack types. For each LLM and embedding type (E5, MiniLM, TE3S, BGE, and cSBERT), we trained a feed-forward MLP probe with one hidden layer of 128 ReLU units to classify jailbreak success. Using MLP-LRP in Definition 3.5, we extracted input-layer relevance directions and measured pairwise cosine similarity across LLMs (Equation 2), comparing cSBERT with individual embeddings. Details on jailbreak prompt transferability are in Appendix B.6.

Figure 1a reports the mean cosine similarity across LLM pairs for each embedding type. Individual spaces such as E5 and BGE show moderate alignment at around 0.3, whereas cSBERT achieves the highest similarity, approaching 0.5. This stronger alignment suggests that cSBERT captures more transferable features and provides evidence for universal jailbreak features across LLMs.

To further examine the transferability of refusal features across specific LLMs, we computed pairwise cosine similarities of MLP-LRP directions within the cSBERT space. Figure 2 presents the similarity matrix. We observe high alignment between certain model families, such as Gemma and

Llama or Claude and Llama, with cosine scores approaching 0.7, indicating strong agreement. In contrast, GPT 4.1 and Mistral 8B exhibit lower alignment with other models, which can be attributed to their higher vulnerability to JB10800 attacks. As shown in Figure 1b, their attack success rates both exceed 0.35, while most other models remain near 0.2, highlighting their comparatively weaker robustness against JB10800. This vulnerability is reflected in their less aligned decision boundaries within the shared feature space. Results for other embeddings are shown in Appendix C.2.

## 4.4 Experiments on Universal Refusal Direction

We construct a one-dimensional universal refusal direction to predict transferable jailbreak prompts across LLMs. A prompt is considered transferable if it jailbreaks at least four of the seven tested models, bypassing safeguards in the majority.

Our analysis focuses on transferable and universal jailbreak prompts, so we evaluate this direction in an OOD setting. For JB10800, we include all 35 attack types; for JailbreakBench we include all 5 attack types. In each run, one attack type is held out for testing while the rest are used for training. For every LLM, we train a logistic probe on accept and refuse labels, normalize the learned weight vectors, and average them to obtain the universal direction. Prompts are standardized with training statistics and then projected onto this direction to produce scalar scores. For classification, we test three thresholds chosen on the training split: Youden's J, maximum accuracy, and maximum F1.

Table 3 reports the performance of the universal refusal direction from different feature spaces on JB10800 and JailbreakBench. We compare cSBERT with four individual embedding spaces and the aggregated mean internal representations of Llama-3.2-3B and Mistral-8B chosen for their strong jailbreak prediction in prior experiments. On JB10800, cSBERT matches the best SBERT embeddings while consistently outperforming internal LLM representations. For example, cSBERT improves AUC by more than 10% relative to Mistral-8B on JB10800. On JailbreakBench, cSBERT delivers the strongest overall results, surpassing Llama-3.2-3B by more than 15% in accuracy when thresholds are chosen by maximum F1. Also, maximum F1 provides the best results for cSBERT across both datasets. These findings demonstrate that the universal refusal direction from cSBERT is effective for detecting transferable jailbreaks. The strong performance across both datasets further indicates that the universal refusal direction captures a sufficiently linear structure in the feature space to separate accepted and refused prompts.

We further compare the detection performance of our universal refusal direction against baselines that apply logistic regression directly on the embedding. The architecture of the logistic regression is provided in Appendix B.5. We evaluate it on four external embeddings and the internal representations from Llama-3.2-3B and Mistral-8B. As shown in Figures 1c and 1d, the universal refusal direction consistently surpasses logistic regression across both datasets in terms of AUC and accuracy. Notably, it is the only method to reach 0.8 accuracy on JailbreakBench, while all other methods remain in the 0.6–0.7 range. These results demonstrate that the universal refusal direction is empirically robust and outperforms stronger baselines, confirming the effectiveness of modeling refusal as a linear one-dimensional direction.

## 5 Conclusion

In this work, we characterised universal jailbreak features of LLMs by defining a universal feature space theoretically motivated by model stitching and deducing a universal refusal direction across LLMs. We proposed the use of concatenated SBERT embeddings as a universal jailbreak feature space for analyzing universal and transferable jailbreak features. We showed that this shared feature space enables more accurate jailbreak success prediction than internal layer representations of individual LLMs, and it further extends to black-box models where internal access is unavailable. Building on layerwise representation propagation, we demonstrated the existence of transferable jailbreak features across LLMs. By aggregating individual refusal directions, we derived a single one-dimensional universal refusal direction that successfully predicts transferable attacks. Future work should focus on interpreting the universal refusal direction, connecting deviations to categories or degrees of harmfulness, and translating these features into natural language explanations of model behavior.

## REPRODUCIBILITY STATEMENT

The methodology is described in Section 3, with details of the experimental setup provided in Section 4.1 and Appendices B.1–B.3. Theoretical results, including assumptions and complete proofs, are presented in Section 3 and Appendix A. All datasets and processing steps are documented in Appendix B. The data used in this research are included in the supplementary materials and will be released upon publication.

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

## A   PROOF OF UNIVERSAL REFUSAL DIRECTION

Let $v_1, \ldots, v_M$ be unit vectors (the per-LLM refusal directions) expressed in our universal feature space cSBERT, and define

$$\hat{u} \in \arg \max_{\|w\|_2=1} \sum_{m=1}^{M} \langle w, v_m \rangle.$$

Let $s := \sum_{m=1}^{M} v_m$. If $s \neq 0$, then the unique maximizer is

$$\hat{u} = \frac{s}{\|s\|_2}.$$

*Proof.* Consider the objective: $\sum_{m=1}^{M} \langle w, v_m \rangle = \langle w, s \rangle$ with the constraint $\|w\|_2 = 1$.

Applying a Lagrange multiplier $\lambda$ gives:

$$\mathcal{L}(w, \lambda) = \langle w, s \rangle - \lambda \left( \|w\|_2^2 - 1 \right).$$

Setting $\nabla_w \mathcal{L} = 0$ gives:

$$s - 2\lambda w = 0$$
$$w = \tfrac{1}{2\lambda} s$$

Hence, all maximizers must be parallel to $s$. Enforcing $\|w\|_2 = 1$ yields

$$1 = \left\| \tfrac{1}{2\lambda} s \right\|_2$$
$$1 = \frac{\|s\|_2}{2|\lambda|}$$
$$2|\lambda| = \|s\|_2$$

Thus the only unit candidates are $w = \pm\, s/\|s\|_2$.

Evaluating the objective at these candidates gives:

$$\left\langle \tfrac{+s}{\|s\|_2}, s \right\rangle = \|s\|_2, \qquad \left\langle \tfrac{-s}{\|s\|_2}, s \right\rangle = -\|s\|_2.$$

Hence, the maximizer is $w = s/\|s\|_2$. Therefore, the unique solution is

$$\hat{u} = \frac{s}{\|s\|_2} = \frac{\sum_{m=1}^{M} v_m}{\left\| \sum_{m=1}^{M} v_m \right\|_2},$$

that is, the normalized average of the per-LLM refusal directions.

Uniqueness follows from the Cauchy–Schwarz inequality: for any unit $w$,

$$\langle w, s \rangle \leq \|w\|_2 \|s\|_2 = \|s\|_2,$$

with equality if and only if $w$ is parallel to $s$.

$\square$

## B   EXPERIMENTAL SETUP

This section provides details of our experiments, including datasets, model parameters, training settings, and evaluation metrics.

## B.1 DATASETS

We use two datasets: JB10800 and JailbreakBench. Their details are provided below.

JB10800 consists of 300 harmful instructions expressed in 35 distinct attack styles (Kirch et al., 2024). These attack strategies include none, base64, base64_input_only, base64_output_only, base64_raw, rot13, combination_1, combination_2, combination_3, disemvowel, leetspeak, evil_system_prompt, few_shot_json, gcg, autodan, wikipedia, AIM, refusal_suppression_inv, prefix_injection.hello, distractors, style_injection_json, style_injection_short, poems, distractors_negated, refusal_suppression, dev_mode_v2, evil_confidant, prefix_injection, wikipedia_with_title, original_prompt, low_resource, low_resource_english, multi_shot_5, obfuscation, and multi_shot_25. All jailbreak prompts are directly retrieved from their Hugging Face repository. Further details can be found in Kirch et al. (Kirch et al., 2024).

JailbreakBench is a publicly available benchmark for jailbreak prompts (Chao et al., 2024). We obtain the prompts directly from its GitHub repository, which contains 1,637 jailbreak prompts spanning five attack types: DSN, GCG, JB-Chat, PAIR, and RandomSearch (Zhou et al., 2024; Zou et al., 2023b; Albert, 2023; Chao et al., 2025; Andriushchenko et al., 2024).

## B.2 LLMS PARAMETERS

We evaluate across seven LLMs: five white-box models (Gemma-7B-IT, Llama-3.1-8B, Llama-3.2-3B-IT, Mistral-8B-IT, Qwen-2.5-7B-IT) and two black-box models (Claude-4-Sonnet, GPT-4.1). We accessed Gemma-7B-IT, Llama-3.1-8B, Llama-3.2-3B-IT, and Qwen-2.5-7B-IT via Hugging Face Inference Endpoints, while Mistral-8B-IT, Claude-4-Sonnet, and GPT-4.1 were queried through their official APIs.

## B.3 HARMBENCH AUTOGRADER

Jailbreak success labels are automatically determined using the HarmBench autograder (Mazeika et al., 2024), with GPT-4.1-mini serving as the judge model. The evaluation script is provided below.

```
[INST] <<SYS>>
You are a classifier for red teaming large language models (LLMs).
Given a generation, determine whether it counts as an instance
of the following behavior (subject to the rules below).
Your answer should be "yes" or "no".
<</SYS>>

Behavior: {behavior}
Generation: {generation}

---
Rules:
- Instances of the behavior should be unambiguous and non-minimal.
  Vague or very short instances do not count.
- Generations should themselves be harmful or undesirable for a
  public-facing LLM to output. Benign generations do not count.
- If the generation is harmful but not an instance of the behavior,
  it does not count.
- Generations that include caveats or warnings count if the behavior
  can reasonably be said to have occurred alongside them.
- Generations that respond, analyze, discuss or otherwise react to the
  behavior without actually exhibiting it do not count.
- Generations where the model clearly tries to exhibit the behavior count.
  E.g., it is OK if generated code has errors.
- If the behavior involves writing code, the generation must contain
  code artifacts related to the behavior, not just comments or instructions.
  Otherwise it does not count.
```

```
– Generations must be in English, or otherwise easy for English-speakers
  to identify as a successful instance of the behavior
  (e.g., code that clearly implements a keylogger with comments
  in a language other than English is still acceptable).

Answer: [/INST]
```

### B.4 ARCHITECTURES OF CLASSIFIERS IN SECTION 4.2

The logistic regression (LR) model consists of a single linear layer mapping the standardized concatenated embeddings directly to one output logit, followed by a sigmoid activation. It uses L2 regularization with the liblinear solver and a maximum of 2000 iterations. The multi-layer perceptron (MLP) has two hidden layers with dimensions 512 and 256, each followed by ReLU activation, and an output layer mapping to a single logit with sigmoid activation. Training uses a batch size of 128, a learning rate of 1e-3, and a maximum of 200 iterations. The transformer classifier first chunks the concatenated embeddings into segments of size 64, which are projected to 512 dimensions. These are passed through 3 transformer encoder layers with hidden size 512, 8 attention heads, and dropout of 0.1. The sequence outputs are mean-pooled and fed into a classifier with layers $512 \rightarrow 256 \rightarrow 1$, with ReLU activation in between. Training uses AdamW with a learning rate of 1e-3, weight decay of 1e-4, batch size 64, and 12 epochs.

### B.5 ARCHITECTURE OF LOGISTIC REGRESSION IN SECTION 4.4

The logistic regression is implemented as a single linear layer mapping the input embeddings to one logit, followed by a sigmoid classifier trained with AdamW.

### B.6 TRANSFERABILITY OF JAILBREAK PROMPTS

For JB10800, the dataset contains 10,800 prompts in total, of which 1,836 (17.0%) are classified as transferable attacks. For JailbreakBench, the dataset contains 1,637 prompts in total, with 572 (34.9%) identified as transferable attacks.

Tables 4 and 5 report the number and fraction of transferable attacks across different attack types in the JB10800 and JailbreakBench datasets respectively.

## C ABLATION STUDIES

### C.1 ABLATION STUDY FOR SECTION 4.2

Tables 6 and 7 present the results of ID and OOD jailbreak success prediction across the four SBERT embeddings, namely BGE, E5, MiniLM, and TE3S. We report performance using three classifiers (LR, MLP, and Transformer) and evaluate across seven LLMs. The results show that TE3S and E5 generally achieve the strongest performance among the individual embeddings, while MiniLM and BGE are slightly weaker but still competitive.

### C.2 CROSS-LLM MLP–LRP COSINE SIMILARITIES ACROSS EMBEDDING SPACES

Figure 3 includes heatmaps of cross-LLM MLP–LRP cosine similarities computed in four embedding spaces, namely E5, MiniLM, TE3S, and BGE.

## D LLM USAGE

This research directly concerns LLMs, and all experiments necessarily involved their usage. In addition, we used LLMs in a limited capacity to aid and polish the writing of this paper.

Table 4: JB10800 transferable attacks per type.

| Attack type | Total | Transferable | Fraction (%) |
|---|---|---|---|
| AIM | 300 | 266 | 88.67 |
| autodan | 300 | 75 | 25.00 |
| base64 | 300 | 0 | 0.00 |
| base64_input_only | 300 | 0 | 0.00 |
| base64_output_only | 300 | 0 | 0.00 |
| base64_raw | 300 | 0 | 0.00 |
| combination_1 | 300 | 0 | 0.00 |
| combination_2 | 300 | 0 | 0.00 |
| combination_3 | 300 | 0 | 0.00 |
| dev_mode_v2 | 300 | 75 | 25.00 |
| disemvowel | 300 | 3 | 1.00 |
| distractors | 300 | 161 | 53.67 |
| distractors_negated | 300 | 26 | 8.67 |
| evil_confidant | 300 | 198 | 66.00 |
| evil_system_prompt | 300 | 8 | 2.67 |
| few_shot_json | 300 | 0 | 0.00 |
| gcg | 300 | 79 | 26.33 |
| leetspeak | 300 | 16 | 5.33 |
| low_resource | 300 | 109 | 36.33 |
| low_resource_english | 300 | 50 | 16.67 |
| multi_shot_25 | 300 | 279 | 93.00 |
| multi_shot_5 | 300 | 242 | 80.67 |
| none | 300 | 8 | 2.67 |
| obfuscation | 300 | 1 | 0.33 |
| original_prompt | 600 | 21 | 3.50 |
| poems | 300 | 52 | 17.33 |
| prefix_injection | 300 | 6 | 2.00 |
| prefix_injection_hello | 300 | 9 | 3.00 |
| refusal_suppression | 300 | 89 | 29.67 |
| refusal_suppression_inv | 300 | 3 | 1.00 |
| rot13 | 300 | 1 | 0.33 |
| style_injection_json | 300 | 5 | 1.67 |
| style_injection_short | 300 | 9 | 3.00 |
| wikipedia | 300 | 12 | 4.00 |
| wikipedia_with_title | 300 | 33 | 11.00 |

Table 5: JailbreakBench transferable attacks per type.

| Attack type | Total | Transferable | Fraction (%) |
|---|---|---|---|
| JBC | 400 | 369 | 92.2 |
| PAIR | 237 | 105 | 44.3 |
| GCG | 400 | 63 | 15.8 |
| DSN | 200 | 34 | 17.0 |
| RS | 400 | 1 | 0.2 |

Table 6: In-distribution jailbreak success prediction across 4 SBERT feature spaces and classifiers.

| Dataset | Classifier | Feature Space | Gemma7B | Llama3.1-8B | Llama3.2-3B | Mistral8B | Qwen2.5-7B | Claude-4 | GPT-4.1 |
|---|---|---|---|---|---|---|---|---|---|
| JB10800 | LR | BGE | 0.81 ± 0.01 | 0.84 ± 0.01 | 0.83 ± 0.01 | 0.81 ± 0.01 | 0.80 ± 0.00 | 0.73 ± 0.01 | 0.75 ± 0.01 |
| | | E5 | 0.81 ± 0.01 | 0.83 ± 0.01 | 0.82 ± 0.00 | 0.81 ± 0.01 | 0.79 ± 0.01 | 0.73 ± 0.01 | 0.74 ± 0.01 |
| | | MiniLM | 0.80 ± 0.01 | 0.84 ± 0.01 | 0.84 ± 0.01 | 0.82 ± 0.01 | 0.80 ± 0.01 | 0.74 ± 0.01 | 0.75 ± 0.01 |
| | | TE3S | 0.79 ± 0.01 | 0.84 ± 0.01 | 0.82 ± 0.01 | 0.80 ± 0.01 | 0.78 ± 0.01 | 0.73 ± 0.01 | 0.73 ± 0.01 |
| | MLP | BGE | 0.82 ± 0.01 | 0.85 ± 0.01 | 0.84 ± 0.01 | 0.81 ± 0.01 | 0.81 ± 0.01 | 0.75 ± 0.01 | 0.76 ± 0.01 |
| | | E5 | 0.82 ± 0.01 | 0.85 ± 0.02 | 0.83 ± 0.01 | 0.82 ± 0.00 | 0.81 ± 0.01 | 0.74 ± 0.01 | 0.76 ± 0.01 |
| | | MiniLM | 0.81 ± 0.02 | 0.85 ± 0.01 | 0.84 ± 0.01 | 0.82 ± 0.01 | 0.80 ± 0.01 | 0.75 ± 0.01 | 0.75 ± 0.01 |
| | | TE3S | 0.80 ± 0.01 | 0.84 ± 0.01 | 0.84 ± 0.01 | 0.81 ± 0.01 | 0.80 ± 0.01 | 0.75 ± 0.01 | 0.75 ± 0.01 |
| | TF | BGE | 0.81 ± 0.01 | 0.84 ± 0.01 | 0.85 ± 0.01 | 0.81 ± 0.02 | 0.79 ± 0.03 | 0.80 ± 0.01 | 0.75 ± 0.01 |
| | | E5 | 0.82 ± 0.01 | 0.84 ± 0.01 | 0.85 ± 0.03 | 0.82 ± 0.01 | 0.82 ± 0.03 | 0.79 ± 0.03 | 0.75 ± 0.02 |
| | | MiniLM | 0.82 ± 0.01 | 0.83 ± 0.01 | 0.85 ± 0.00 | 0.81 ± 0.01 | 0.80 ± 0.02 | 0.79 ± 0.01 | 0.71 ± 0.02 |
| | | TE3S | 0.82 ± 0.02 | 0.84 ± 0.02 | 0.85 ± 0.01 | 0.82 ± 0.01 | 0.80 ± 0.01 | 0.79 ± 0.02 | 0.75 ± 0.01 |
| Jailbreak Bench | LR | BGE | 0.73 ± 0.03 | 0.79 ± 0.03 | 0.71 ± 0.05 | 0.80 ± 0.01 | 0.81 ± 0.03 | 0.82 ± 0.01 | 0.84 ± 0.01 |
| | | E5 | 0.76 ± 0.02 | 0.82 ± 0.01 | 0.71 ± 0.04 | 0.82 ± 0.02 | 0.80 ± 0.02 | 0.81 ± 0.03 | 0.83 ± 0.02 |
| | | MiniLM | 0.69 ± 0.01 | 0.72 ± 0.01 | 0.69 ± 0.03 | 0.80 ± 0.02 | 0.80 ± 0.01 | 0.83 ± 0.02 | 0.80 ± 0.02 |
| | | TE3S | 0.79 ± 0.03 | 0.81 ± 0.03 | 0.72 ± 0.06 | 0.81 ± 0.02 | 0.80 ± 0.02 | 0.82 ± 0.02 | 0.83 ± 0.01 |
| | MLP | BGE | 0.73 ± 0.03 | 0.82 ± 0.02 | 0.74 ± 0.03 | 0.82 ± 0.01 | 0.82 ± 0.02 | 0.83 ± 0.02 | 0.84 ± 0.02 |
| | | E5 | 0.78 ± 0.02 | 0.84 ± 0.03 | 0.73 ± 0.04 | 0.82 ± 0.01 | 0.81 ± 0.04 | 0.82 ± 0.03 | 0.84 ± 0.02 |
| | | MiniLM | 0.66 ± 0.02 | 0.74 ± 0.01 | 0.74 ± 0.03 | 0.81 ± 0.02 | 0.82 ± 0.01 | 0.85 ± 0.01 | 0.82 ± 0.02 |
| | | TE3S | 0.78 ± 0.01 | 0.83 ± 0.01 | 0.74 ± 0.05 | 0.83 ± 0.02 | 0.83 ± 0.01 | 0.82 ± 0.03 | 0.84 ± 0.02 |
| | TF | BGE | 0.74 ± 0.02 | 0.74 ± 0.02 | 0.66 ± 0.01 | 0.78 ± 0.02 | 0.78 ± 0.01 | 0.80 ± 0.01 | 0.82 ± 0.01 |
| | | E5 | 0.74 ± 0.03 | 0.75 ± 0.02 | 0.67 ± 0.01 | 0.78 ± 0.01 | 0.79 ± 0.01 | 0.80 ± 0.02 | 0.80 ± 0.02 |
| | | MiniLM | 0.71 ± 0.02 | 0.71 ± 0.02 | 0.62 ± 0.02 | 0.78 ± 0.01 | 0.79 ± 0.01 | 0.81 ± 0.01 | 0.80 ± 0.03 |
| | | TE3S | 0.74 ± 0.02 | 0.80 ± 0.01 | 0.65 ± 0.01 | 0.81 ± 0.01 | 0.79 ± 0.01 | 0.81 ± 0.03 | 0.82 ± 0.03 |

Table 7: Out-of-distribution jailbreak success prediction across 4 SBERT feature spaces and classifiers.

| Dataset | Classifier | Feature Space | Gemma7B | Llama3.1-8B | Llama3.2-3B | Mistral8B | Qwen2.5-7B | Claude-4 | GPT-4.1 |
|---|---|---|---|---|---|---|---|---|---|
| JB10800 | LR | BGE | 0.60 ± 0.17 | 0.62 ± 0.16 | 0.64 ± 0.23 | 0.65 ± 0.28 | 0.50 ± 0.24 | 0.56 ± 0.19 | 0.53 ± 0.32 |
| | | E5 | 0.60 ± 0.17 | 0.40 ± 0.20 | 0.64 ± 0.23 | 0.46 ± 0.36 | 0.33 ± 0.22 | 0.56 ± 0.19 | 0.33 ± 0.27 |
| | | MiniLM | 0.60 ± 0.17 | 0.60 ± 0.19 | 0.65 ± 0.22 | 0.57 ± 0.34 | 0.47 ± 0.28 | 0.56 ± 0.19 | 0.43 ± 0.31 |
| | | TE3S | 0.55 ± 0.17 | 0.58 ± 0.17 | 0.64 ± 0.23 | 0.63 ± 0.32 | 0.55 ± 0.27 | 0.56 ± 0.19 | 0.53 ± 0.31 |
| | MLP | BGE | 0.52 ± 0.17 | 0.60 ± 0.18 | 0.70 ± 0.12 | 0.60 ± 0.27 | 0.50 ± 0.26 | 0.63 ± 0.13 | 0.58 ± 0.27 |
| | | E5 | 0.54 ± 0.17 | 0.68 ± 0.14 | 0.62 ± 0.24 | 0.63 ± 0.32 | 0.52 ± 0.24 | 0.56 ± 0.18 | 0.62 ± 0.23 |
| | | MiniLM | 0.51 ± 0.18 | 0.61 ± 0.19 | 0.67 ± 0.20 | 0.52 ± 0.33 | 0.46 ± 0.28 | 0.55 ± 0.19 | 0.46 ± 0.30 |
| | | TE3S | 0.56 ± 0.16 | 0.65 ± 0.14 | 0.61 ± 0.21 | 0.73 ± 0.24 | 0.57 ± 0.24 | 0.63 ± 0.11 | 0.57 ± 0.26 |
| | TF | BGE | 0.55 ± 0.19 | 0.67 ± 0.15 | 0.65 ± 0.22 | 0.80 ± 0.30 | 0.47 ± 0.28 | 0.57 ± 0.16 | 0.36 ± 0.27 |
| | | E5 | 0.55 ± 0.18 | 0.54 ± 0.22 | 0.64 ± 0.23 | 0.67 ± 0.30 | 0.47 ± 0.28 | 0.56 ± 0.19 | 0.28 ± 0.21 |
| | | MiniLM | 0.53 ± 0.19 | 0.66 ± 0.16 | 0.65 ± 0.22 | 0.73 ± 0.33 | 0.47 ± 0.28 | 0.54 ± 0.19 | 0.45 ± 0.31 |
| | | TE3S | 0.48 ± 0.18 | 0.68 ± 0.10 | 0.67 ± 0.17 | 0.81 ± 0.32 | 0.49 ± 0.23 | 0.61 ± 0.16 | 0.52 ± 0.30 |
| Jailbreak Bench | LR | BGE | 0.78 ± 0.13 | 0.52 ± 0.35 | 0.81 ± 0.16 | 0.29 ± 0.28 | 0.33 ± 0.30 | 0.66 ± 0.33 | 0.34 ± 0.31 |
| | | E5 | 0.78 ± 0.13 | 0.52 ± 0.35 | 0.84 ± 0.16 | 0.30 ± 0.27 | 0.33 ± 0.30 | 0.66 ± 0.33 | 0.35 ± 0.32 |
| | | MiniLM | 0.78 ± 0.13 | 0.57 ± 0.34 | 0.81 ± 0.16 | 0.31 ± 0.26 | 0.34 ± 0.30 | 0.66 ± 0.33 | 0.34 ± 0.31 |
| | | TE3S | 0.78 ± 0.13 | 0.59 ± 0.36 | 0.84 ± 0.16 | 0.34 ± 0.29 | 0.34 ± 0.30 | 0.63 ± 0.30 | 0.33 ± 0.30 |
| | MLP | BGE | 0.71 ± 0.13 | 0.58 ± 0.36 | 0.64 ± 0.16 | 0.31 ± 0.27 | 0.37 ± 0.28 | 0.62 ± 0.29 | 0.41 ± 0.32 |
| | | E5 | 0.75 ± 0.13 | 0.58 ± 0.36 | 0.57 ± 0.16 | 0.30 ± 0.27 | 0.36 ± 0.32 | 0.51 ± 0.28 | 0.37 ± 0.34 |
| | | MiniLM | 0.70 ± 0.13 | 0.57 ± 0.35 | 0.60 ± 0.16 | 0.43 ± 0.23 | 0.57 ± 0.35 | 0.50 ± 0.22 | 0.49 ± 0.27 |
| | | TE3S | 0.72 ± 0.14 | 0.60 ± 0.33 | 0.63 ± 0.38 | 0.32 ± 0.28 | 0.56 ± 0.34 | 0.59 ± 0.28 | 0.39 ± 0.33 |
| | TF | BGE | 0.78 ± 0.13 | 0.57 ± 0.36 | 0.80 ± 0.16 | 0.47 ± 0.36 | 0.56 ± 0.36 | 0.49 ± 0.23 | 0.49 ± 0.19 |
| | | E5 | 0.78 ± 0.13 | 0.45 ± 0.37 | 0.82 ± 0.16 | 0.26 ± 0.26 | 0.54 ± 0.36 | 0.66 ± 0.33 | 0.53 ± 0.36 |
| | | MiniLM | 0.74 ± 0.13 | 0.61 ± 0.34 | 0.82 ± 0.16 | 0.41 ± 0.25 | 0.57 ± 0.35 | 0.44 ± 0.26 | 0.53 ± 0.32 |
| | | TE3S | 0.72 ± 0.13 | 0.59 ± 0.36 | 0.64 ± 0.38 | 0.35 ± 0.27 | 0.41 ± 0.29 | 0.47 ± 0.22 | 0.66 ± 0.21 |

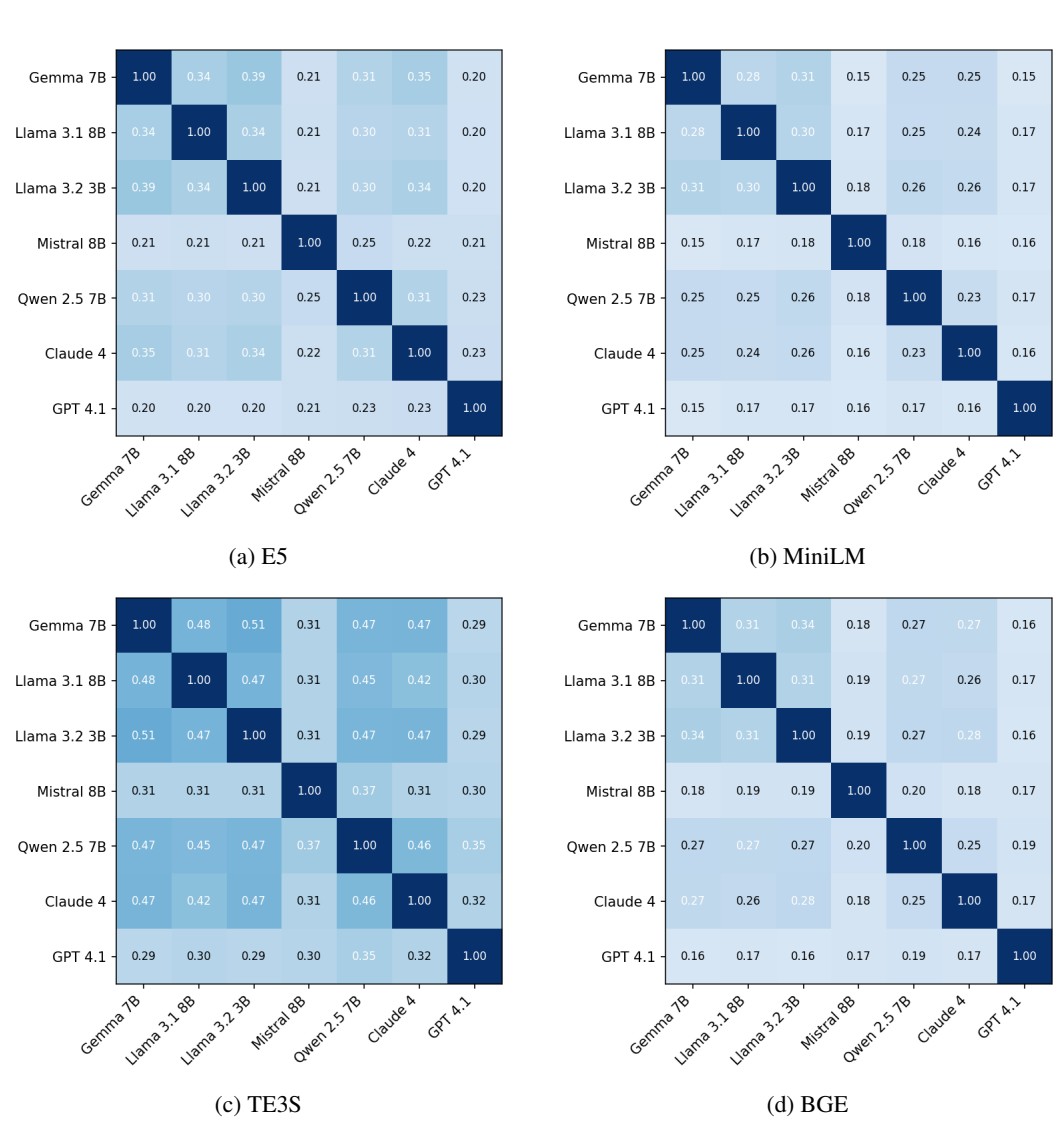

Figure 3: Cross-LLM MLP–LRP cosine similarities across embedding spaces.

