# OpenReview forum: "Characterising Universal Jailbreak Features and Refusal Direction in LLMs"
_ICLR.cc/2026/Conference — ICLR 2026 Conference Withdrawn Submission_

### Official Review · Reviewer_QGj8 · 2025-10-27

**Soundness:** 2
**Presentation:** 2
**Contribution:** 2
**Rating:** 4
**Confidence:** 4

**Summary:**

This paper considers the hypotheses that different language models develop similar and linear representations of features corresponding to refusal and jailbreaks. Based on concepts from model stitching, this paper develops a method for embedding the internal representations of many models into a single shared feature space. In turn, this paper does experiments to show that that feature space allows for successful probing and probation with relatively high generality. Based on all of this evidence together, the paper argues that models represent refusal and jailbreak features using universal linear features.

**Strengths:**

S1: I am not super familiar with related methods, but from my reading, I think that the stitching penalty and the concept of predictive sufficiency are clever.

S2: I think the method worked better than I thought it would. I'm updating a bit toward thinking that there might be something interesting or worthwhile about considering the idea of universal jailbreak features that are approximately linear.

**Weaknesses:**

W1: the last two sentences of a "refusal directions" paragraph don't seem to make a valid point to me. If this paper is going to argue based on an analysis of multiple models, that refusal is approximately linear, then past evidence that it is not linear cannot be dismissed because it focused on one model at a time. Focusing on one model at a time versus multiple models is independent of whether or not a feature is genuinely linear.

W2: related to the last point, I think this might be two things. One is the ability of a behavior to be modulated by a linear intervention. The other is that feature being "genuinely" linear. I think this claims the second, but only shows the first. To also show the second, this paper would need to show that the linear method does as well or better than comparable nonlinear ones.

W3: On one hand, definitions 3.4 and 3.6 are reasonable. But on the other hand, the paper does a lot of work by definition. The claims about universal jailbreak features being connected to this definition is less meaningful than being connected to a practical property or capacity of language models.

W4: Tables 1 and 2 make me wonder: "Well cSBERT representations of a models in bedding can not contain any more information about the models, and then the embedding themselves. So any improvements in probing using those representations must be based on how convenient it is to fit features in cSBERT feature space. But I don't see how the convenience of fitting features in a space is connected to the Central claims of this paper about linear and universal features."

W5: Hm, in table 2, I am not surprised that most values are well above zero given that the cSBERT Space was created precisely to align the representations of all the models. There's not really a way of easily grounding our interpretation of the specific numbers in the table. I'm not inclined to assume that seeing many cosine similarities less than 0.5 is extremely impressive.

Minor
- Fig 1 could be cleaned up. Use plt.subplots instead of making separate plots
- This might be partially my fault, but I did not find this paper to be the clearest that I have ever read.

**Questions:**

Q1: I am not understanding fully what IntRep is based on Kirch et al? Does it just refer to the neural activations in the target model? "Intrep" does not appear in the paper.

Q2: I am not fully understanding why MLP and tensor probes are used in tables one and two. If the paper is arguing about the linearity of jailbreak features, wouldn't successful MLP/transformer probes in the cSBERT space not support that?

Q3: sorry, I am not understanding how the methodology in section 3.1 works for blackbox models.

---

> ### Author Response · Authors · 2025-12-01
>
> We appreciate the reviewer’s recognition of the novelty and contributions of our methodology. We are also grateful for the constructive and thoughtful feedback, which has been genuinely helpful in clarifying our ideas and guiding several concrete improvements in the revised manuscript.
>
> We would like to clarify that the jailbreak features we identify exhibit linear structure within the cSBERT universal feature space that we construct. We agree that this point requires more explicit exposition in the paper, and we will improve the explanation. We also recognise that the connection between the mathematical definitions, the proofs, and the high-level overview can be strengthened. Moreover, we plan to conduct additional experiments to further validate the applicability of cSBERT for discovering universal refusal directions across models.
>
> Regarding the concern about IntRep, Kirch et al. propose analysing the internal representations of target models by aggregating layer-wise activations using mean pooling. Because their paper defines this internal-representation feature space and provides the corresponding dataset (which they did not name as well, so we call it JB10800), we refer to it simply as IntRep. We will add a more detailed explanation of this in the revised version.
>
> To address concerns about using MLP and Transformer classifiers, we include these baselines in Tables 1 and 2 to enable a fair comparison with prior work and to demonstrate that the cSBERT representation is sufficient for predicting jailbreak success. Finally, on applicability to black-box models, our main claim is that both open-source and black-box systems share universal jailbreak features, which can be jointly characterised through a linear, one-dimensional universal refusal direction.

---

### Official Review · Reviewer_tE23 · 2025-10-27

**Soundness:** 3
**Presentation:** 3
**Contribution:** 2
**Rating:** 2
**Confidence:** 4

**Summary:**

This paper examines the internal representation of jailbreaks and the refusal directions within LLMs. Many prior works examine LLMs individually, while this paper considers if there is a universality to the refusal directions shared across LLMS. The authors show a shared structure between several LLMs, and creates single dimensional refusal direction which predicts jailbreaks transfer across LLMs.

**Strengths:**

This paper addresses an interesting problem, as more grounded and generalizable methods to analyse intrinsic LLM properties are needed.

The experiments cover a large number of models and datasets, empirically testing the proposed methods in several scenarios including OOD datasets, black box proprietary LLMS, cross family models (Gemma, Llama, Mistral, Qwen) etc.

**Weaknesses:**

Parts of section 3 are difficult to understand in terms of writing and motivation for various steps, with perhaps over-dense mathematical decoration.  Thus, potentially I may have missed some aspects around it; for example, why averaging to a single dimensionality would be theoretically motivated to give universality. That there are common feature spaces within similar ML models is well known - and if a jailbreak is likely to work on a vary strong model, then it is likely to transfer to a weaker model effectively - thus heuristically it makes sense and is a known phenomenon.

GPT-4.1-mini is used as the Harmbench autograder, while also being one of the model's evaluated in Tables 1 and 2, thus potentially biasing its results. Also transfer between the Llama models seems likely seeing how closely related they would be.

The results in Tables 1 and 2 do show improvement over the baseline, although the delta seem modest. This is particularly pronounced in Table 2 with the variance on the results often being several times the size of any difference between methods.

Perhaps conceptually it's unclear where this paper lies: it does not seem to perform a through analysis of what these universal directions represent semantically or in actionable terms, limiting its contributions there, yet is also limited in a security context as it does not directly aim to boost the robustness of a model to jailbreak attacks. Generally, it's unclear how significant from a security perspective predicting jailbreak transfer effectiveness actually is.

**Questions:**

No specific queries.

---

> ### Author Response · Authors · 2025-12-01
>
> We thank the reviewer for recognising the research gap addressed by our work and for acknowledging the breadth of the empirical results. We agree that the linkage between the mathematical components and the high-level conceptual overview can be strengthened.
>
> We also appreciate the suggestion regarding the HarmBench autograder. Using an alternative autograder would indeed enhance the robustness of our experimental results, and we plan to incorporate an additional set of evaluations in the revised version. Furthermore, we intend to refine our methodology (potentially through parameter tuning) to reduce variance and improve stability. Finally, we will expand our analysis to more thoroughly examine the interpretation and implications of the universal refusal direction.

---

### Official Review · Reviewer_ZR1c · 2025-10-29

**Soundness:** 2
**Presentation:** 2
**Contribution:** 3
**Rating:** 4
**Confidence:** 3

**Summary:**

This paper focuses on the transferability of adversarial examples in LLMs and presents a way to extract universal jailbreak features. To be detailed, the script proved that the mean direction of the model-level rejection direction is the optimal universal refusal direction. In experiments, the aggregated rejection direction performs better than directions derived directly from internal representations under ID and OOD settings, demonstrating the effectiveness of such a method.

**Strengths:**

- The problem is highly valued. The capture of universal jailbreak features could contribute to the understanding of model behavior when faced with adversarial inputs.
- The theoretical framework for deriving the universal feature space from pair-wise stitching is solid.

**Weaknesses:**

My biggest concern is that the paper is not easy to follow, and maybe more explanations are required for engineering-background readers. For a method with complicated parts, I may find it important to explain the necessity of the introduction of different components, for example, why it is required, and how the proof provides support to the block. For example, in Section 3.1, three definitions are established to clarify the model stitching, but finally, the universal space is simply constructed by the concatenation of embeddings from multiple encoders. No validation is presented to demonstrate that the proposed feature space fulfills the predictive efficiency, and there is also no optimization process on the defined stitching penalty to get the space. Without the theoretical framework, it seems that combining the embeddings from multiple encoders to get a universal space still makes sense.

Besides, I also have doubts about the modelling of universal refusal direction. It makes sense, even without the creation of the argmax problem, that averaging the direction of model-wise refusal direction is reasonable to have better transferability than using the direction from one simple model. The question is, there is no theoretical analysis to ensure such an idea. And we still need experiments to check whether the mean vector, even if it is the optimal solution to Proposition 1, has a better performance.

In summary, I am not fully convinced by the design of such a complicated method, or, to be more accurate, the theoretical stuff. The current version does not seem natural to me. Please correct me if I miss some points or make anything misunderstood.

**Questions:**

- What is the relationship between refusal direction and jailbreak features? Do you mean that after getting the universal space, we could calculate the jailbreak features for each model-instruction pair, and predict the success of the jailbreak attack by the universal refusal direction (which is model-agnostic)?
- How to determine the layers on which to calculate the universal space and jailbreak features?
- How could such a method work on commercial models? Do you indicate that the space calculated on these models, and the refusal direction derived from open-source models, transfers well to the prediction of jailbreak ASR?

---

> ### Author Response · Authors · 2025-12-01
>
> We acknowledge that there is room for improvement in the paper, particularly in clarifying the connection between the model-stitching formulation and the practical use of concatenated embeddings from multiple encoders. We also agree that additional empirical evidence would help further support the theoretical foundations of our method.
>
> Regarding the relationship between refusal directions and jailbreak features: the reviewer’s summary is correct. Our pipeline first constructs the universal feature space, then identifies transferable jailbreak features, and finally derives the model-agnostic universal refusal direction, which we use to predict jailbreak success.
>
> On the question about the “layer” or representation used to compute the universal space and jailbreak features, we are unsure what is being referred to. Our universal feature space is built directly from sentence-embedding representations (cSBERT), from which we also extract jailbreak features. If the reviewer is instead referring to the IntRep feature space introduced in Kirch et al. [1], we followed the authors’ instructions as described in their paper.
>
> Finally, concerning applicability to commercial models: we found that the universal refusal direction derived in the cSBERT space generalises well to both open-source and black-box systems, as shown in our results on Claude-4 and GPT-4.1.
>
> [1] Nathalie Kirch, Constantin Weisser, Severin Field, Helen Yannakoudakis, and Stephen Casper. What features in prompts jailbreak LLMs? Investigating the mechanisms behind attacks. arXiv preprint arXiv:2411.03343, 2024.

---

### Official Review · Reviewer_49Fr · 2025-10-31

**Soundness:** 2
**Presentation:** 2
**Contribution:** 3
**Rating:** 4
**Confidence:** 3

**Summary:**

This paper explores the possibility that large language models (LLMs) share common safety-related representations, proposing that there exist universal jailbreak features and a universal refusal direction underlying their refusal behavior. To investigate this, the authors construct a universal feature space (cSBERT) by concatenating multiple SBERT embeddings, theoretically grounded in model stitching. They then employ an MLP-based Layer-wise Relevance Propagation (MLP-LRP) approach to identify transferable jailbreak features within this space, and derive a one-dimensional universal refusal direction by averaging refusal vectors learned from multiple LLMs. Experiments on two benchmarks across seven models show that cSBERT improves jailbreak detection by around 10% and that the universal refusal direction achieves approximately 0.8 accuracy for transferable attack detection.

**Strengths:**

1. Novel Concept and Goal. The central idea of uncovering universal, transferable refusal features across different LLMs is original, addressing an underexplored area of cross-model safety analysis.
2. Model-Agnostic and Black-Box Applicability. The framework does not rely on internal activations, making it applicable to both open and proprietary models, which enhances its practical relevance.
3. Theoretical Motivation via Model Stitching. Extending the concept of model stitching to a shared universal feature space is conceptually appealing and provides a plausible theoretical foundation.

**Weaknesses:**

1. Writing and Presentation Issues. The paper suffers from unclear exposition and poor structural organization. Many sections dive deeply into mathematical and procedural details without first establishing a clear high-level overview. Key ideas such as how the universal feature space connects to model stitching or how the universal direction is computed, are obscured by dense notation and long formal derivations.
2. Weak Theoretical–Empirical Connection. The model stitching formulation (Definitions 3.1–3.3) is mathematically sound but not tightly connected to the empirical experiments. In practice, the authors simply concatenate SBERT embeddings and train classifiers, which feels ad hoc compared to the theory.
3. Limited Interpretability and Lack of Insight. The universal refusal direction is treated as a black-box predictor with little discussion of what linguistic, semantic, or safety properties it captures. Without qualitative analysis, examples, or visualization, the results remain difficult to interpret.
4. Lack of Analytical Depth in Experiments. The experimental section mainly reports numerical accuracy and AUC values without deeper analysis of what these results reveal about the proposed methods. There is no exploration of the characteristics or limitations of the universal feature space or refusal direction, such as robustness, direction interpretability, or relation to specific attack types. This limits the scientific insight of the empirical study.

**Questions:**

1. The paper assumes that refusal behaviors across different LLMs share a universal linear structure. Could you provide empirical or theoretical justification to validate this?
2. Why is simple concatenation of multiple SBERT embeddings (cSBERT) the most appropriate way to build a universal feature space? Have you tested other fusion strategies to verify that concatenation truly captures cross-model semantics?
3. Since cSBERT relies heavily on sentence-embedding models trained on specific corpora, to what extent might the observed universality be an artifact of these embeddings rather than an intrinsic property of LLMs?
4. Can you provide qualitative examples or linguistic analyses showing how prompts that project strongly or weakly onto this direction differ semantically? For instance, do they differ in tone, harmfulness, or instruction framing?

---

> ### Author Response · Authors · 2025-12-01
>
> Thank you for your appreciation of the novelty of our work. We agree that the paper would benefit from a more straightforward high-level overview before presenting the more technical mathematical components.
>
> Regarding the universal linear structure, we do not assume this property; instead, we demonstrate and validate it empirically in Section 4. We also do not claim that cSBERT is the “most appropriate” method for constructing a universal feature space. Our contribution is to show—for the first time, to the best of our knowledge—that such a universal feature space can be constructed and used to investigate whether a shared refusal direction exists across models. This contribution is also recognised by other reviewers.
>
> To address concerns that the observed universality may be an artifact of the chosen embeddings rather than an intrinsic property of LLMs, we would like to clarify that our core claim is conditional: given the universal feature space we propose, the refusal component shared across models lies in a one-dimensional linear direction. This is the property we formally analyse and empirically support throughout the paper.
>
> Finally, we agree that further analysis on prompts that project strongly or weakly onto this universal refusal direction would strengthen the exposition, and we appreciate the reviewer’s suggestion.

---

### Official Review · Reviewer_aTCL · 2025-11-01

**Soundness:** 2
**Presentation:** 3
**Contribution:** 2
**Rating:** 2
**Confidence:** 5

**Summary:**

The paper discusses a method for identifying universal jailbreak features across different LLMs. It is focusing on shared features psace. and investigating a universal refusal direction by averaging over refusal vectors. The authors say their methos enables jailbreak detection across models, including black box models. There is theorectical motivation provided (model stitching). Empirical validation includes two jailbreak datasets and 7 model.s

**Strengths:**

- paper moves existing analyses beyonde single models. This is a contribution.
- paper uses external embeddings and allows for black-box application. important for practice
- Mdel stiching is an interesting motivation
. the evaluation is very comprehensive

**Weaknesses:**

The paper places itself incorrectly into the existing literature. The sentence  More recent analyses further highlighted the features exploited by jailbreak prompts, suggesting they often leverage non-universal and nonlinear properties to circumvent
safeguards (Kirch et al., 2024; Ball et al., 2024; Han et al., 2025). is incorrect and shows that the existing literature was not carefullly considered. The cited literature already showed that jailbreak features are transferrable and that there is geometric similarity across vectors. Even the equations used here are similar to those of the cited literature when discussion harmfulness supression and linear refusal direction.  The paper would be stronger if it would place itself as an extension of the past findings rather than a seemingly contradition and focus on its contributions which are cross-model tranferability (not just cross jailbreak type), that there is an external not an internal embedding space, and that they have a formal model stitching framework.

**Questions:**

Is refusal linear or nonlinear? Your paper simultaneously cites evidence for multidimensional/nonlinear refusal (line 97) and claims one-dimensional/linear refusal (Section 3.3, Proposition 1). Which is correct? If both are true, how?
If it's nonlinear in internal spaces but linear in cSBERT, why?
Is their "universal refusal direction" a mechanistic claim or just a detection heuristic?
Is a jailbreak that works on all open-source models but no proprietary models (or vice versa) "transferable" in a meaningful sense?
How does this threshold relate to actual threat models?
Section 5.4 finds inconsistent relationship between harmfulness reduction and ASR. Does this falsify harmfulness suppression as the mechanism, or is your measurement inadequate? If multiple mechanisms exist, what are they?
Why is 4/7 models the threshold for "transferable"? How do results change with other thresholds (3/7, 5/7)?

---

> ### Author Response · Authors · 2025-12-01
>
> Thank you for the thoughtful comments. We agree that the positioning in the related-work section can be strengthened, and we acknowledge that some of the phrasing in the current draft may unintentionally suggest a contradiction rather than a continuation of prior work.
>
> Regarding the discussion on refusal directions, our intention in Line 97 was to summarise the existing debate in the literature. As noted in Lines 91–98, some works argue that refusal behaviour may be nonlinear or multidimensional, while others demonstrate strong linearity. This passage appears solely as a literature review, not as our own claim. Our actual claim is made later in Section 3.3 (Proposition 1): in the shared external cSBERT space, the universal refusal component across models exhibits a one-dimensional linear direction. This is the property we formally analyse and empirically validate throughout the paper.
>
> On the reviewer’s concern about “Section 5.4,” we assume this refers to Section 4.4, as the paper contains no Section 5.4. However, we are unsure which results the comment refers to, since our experiments do not evaluate or report any form of harmfulness reduction. Across all experiments, our objective is strictly to predict jailbreak success. We do not attempt to reduce harmfulness, intervene in model behaviour, or perform any safety-oriented steering. Our framing is focused purely on understanding and predicting which prompts bypass refusal.

---

### Note · Authors · 2025-12-01

**Comment:**

We would like to sincerely thank all reviewers for their time and constructive feedback. Despite the withdrawal of our submission, we have addressed each reviewer’s concerns below and gratefully acknowledge their thoughtful and constructive comments on our work.

**Withdrawal Confirmation:**

I have read and agree with the venue's withdrawal policy on behalf of myself and my co-authors.